# Suppressive Role of ACVR1/ALK2 in Basal and TGFβ1-Induced Cell Migration in Pancreatic Ductal Adenocarcinoma Cells and Identification of a Self-Perpetuating Autoregulatory Loop Involving the Small GTPase RAC1b

**DOI:** 10.3390/biomedicines10102640

**Published:** 2022-10-20

**Authors:** Hendrik Ungefroren, Rüdiger Braun, Olha Lapshyna, Björn Konukiewitz, Ulrich F. Wellner, Hendrik Lehnert, Jens-Uwe Marquardt

**Affiliations:** 1First Department of Medicine, University Hospital Schleswig-Holstein, Campus Lübeck, D-23538 Lübeck, Germany; 2Institute of Pathology, University Hospital Schleswig-Holstein, Campus Kiel, D-24105 Kiel, Germany; 3Clinic for Surgery, University Hospital Schleswig-Holstein, Campus Lübeck, D-23538 Lübeck, Germany; 4University of Salzburg, 5020 Salzburg, Austria

**Keywords:** ALK2, ALK5, epithelial subtype, invasion, mesenchymal, phenotype, migration, pancreatic cancer, RAC1, RAC1

## Abstract

Pancreatic ductal adenocarcinoma (PDAC) cells are known for their high invasive/metastatic potential, which is regulated in part by the transforming growth factor β1 (TGFβ1). The involvement of at least two type I receptors, ALK5 and ALK2, that transmit downstream signals of the TGFβ via different Smad proteins, SMAD2/3 and SMAD1/5, respectively, poses the issue of their relative contribution in regulating cell motility. Real-time cell migration assays revealed that the selective inhibition of ALK2 by RNAi or dominant-negative interference with a kinase-dead mutant (ALK2-K233R) strongly enhanced the cells’ migratory activity in the absence or presence of TGFβ1 stimulation. Ectopic ALK2-K233R expression was associated with an increase in the protein levels of RAC1 and its alternatively spliced isoform, RAC1b, both of which are implicated in driving cell migration and invasion. Conversely, the RNAi-mediated knockdown or CRISPR/Cas9-mediated knockout of RAC1b resulted in the upregulation of the expression of ALK2, but not that of the related BMP type I receptors, ALK3 or ALK6, and elevated the phosphorylation of SMAD1/5. PDAC is a heterogeneous disease encompassing tumors with different histomorphological subtypes, ranging from epithelial/classical to extremely mesenchymal. Upon treatment of various established and primary PDAC cell lines representing these subtypes with the ALK2 inhibitor, LDN-193189, well-differentiated, epithelial cell lines responded with a much stronger increase in the basal and TGFβ1-dependent migratory activity than poorly differentiated, mesenchymal ones. These data show that (i) ALK2 inhibits migration by suppressing RAC1/RAC1b proteins, (ii) ALK2 and RAC1b act together in a self-perpetuating the autoregulatory negative feedback loop to mutually control their expression, and (iii) the ALK2 antimigratory function appears to be particularly crucial in protecting epithelial subtype cells from becoming invasive, both spontaneously and in a TGFβ-rich tumor microenvironment.

## 1. Introduction

Ninety percent of cancer deaths are caused by metastasis; however, the pathogenesis and mechanisms underlying this event are still poorly understood. Metastasis is believed to consist of four distinct steps: invasion, intravasation, extravasation, and metastatic colonization. The first step, the acquisition of invasive ability and motility, is the rate-limiting step in the metastatic cascade [1] and occurs as part of the epithelial–mesenchymal transition (EMT), a developmental program highjacked by cancer cells. Controlling this initial step of metastasis is crucial for the development of novel strategies to prevent cancer metastasis [2]. The final step in the cascade, metastatic colonization, requires the reverse process, the mesenchymal–epithelial transition (MET), to allow the cells to terminate an invasion and become proliferative to support the metastatic outgrowth [1]. Hence, the timely application of EMT or MET inhibitors may prevent the generation of invasive cells, or large organ-damaging metastases, respectively.

A crucial driver of the EMT and associated cell invasion is the TGFβ, which also implies that inhibiting the TGFβ can prevent the EMT and eventually promote the MET. An important class of factors that acts as natural antagonists of the TGFβ in cancer progression are the bone morphogenetic proteins (BMPs) [3]. For instance, exogenous BMP4 inhibited the TGFβ-induced EMT and epithelial marker downregulation, as well as mesenchymal marker upregulation in retinal pigment epithelium cells [4]. In addition, BMP4 treatment attenuated the TGFβ-induced gel contraction, cell migration, and Smad2/3 phosphorylation [4], while BMP7 is known to counteract the TGFβ1-induced EMT in the developmental stages [5,6].

Signals induced by the TGFβ or BMPs are transmitted by type I serine/threonine kinase receptors. While the TGFβ utilizes the type I receptor activin receptor-like kinase 5 (ALK5) with the subsequent phosphorylation of SMAD2/3, the BMPs use either the activin type I receptor ALK2 (also termed ACVR1, encoded by *ACVR1*), ALK3, or ALK6, all of which phosphorylate SMAD1/5. All four receptor-regulated Smad proteins, after their C-terminal phosphorylation, associate with SMAD4 to be translocated to the nucleus where they promote or repress the transcriptional activity of TGFβ target genes [7]. ALK2 mediates responses to BMP7 [7], and many findings suggest that BMP7/ALK2/SMAD1/5 signaling can induce MET via the inhibition of EMT-associated transcription factors, such as Slug, Twist-1, and Snail [8,9] and reduces the invasion and migration ability. Likewise, in pancreatic cancer cells, MET phenotypes were triggered by NGD16 through the Par-4-dependent augmentation of ALK2/Smad4 signaling [10]. Due to the MET promoting and invasion inhibiting ability of BMP7 or ALK2, BMP7/ALK2 signaling has the potential to act as a metastasis inhibitor, i.e., in human melanoma cells [8].

While in response to BMP7 binding, ALK2-SMAD1/5 activation induces the MET [8,9], or antagonizes the TGFβ-induced EMT [5,6], ALK2 has also been reported to be directly involved in the TGFβ-induced C-terminal phosphorylation of SMAD1 and SMAD5, and SMAD1/5-dependent cellular responses [11,12]. Subsequently, it was demonstrated that in contrast to the TGFβ-induced SMAD2/3 phosphorylation, the TGFβ-induced SMAD1/5 phosphorylation required ALK2, in addition to ALK5 [11]. After the initial phosphorylation of ALK5, ALK5 phosphorylates and thereby activates ALK2, which then phosphorylates SMAD1/5. Moreover, combinatorial signaling via both the SMAD2/3 and SMAD1/5 pathways turned out to be essential for the full TGFβ-induced transcriptional program and physiological responses, such as the TGFβ-induced EMT in NMuMG and EpRas cells [11]. From the studies cited above, it appears that ALK2 can either promote or inhibit the EMT (in which case it will promote the MET), depending on the stimulatory ligand and its ability to be activated by ALK5.

Given the central role of ALK5 in transmitting the TGFβ signal, the agents that modulate its expression or activation are critical determinants of the cells’ sensitivity to this growth factor. We have recently identified RAC1b, a splice isoform of human *RAC1*, as an important negative regulator of TGFβ signaling in PDAC-derived cells by its ability to suppress the expression of ALK5 and, as a consequence, the activation of SMAD3 [13]. Since, conversely, RAC1b protein levels are downregulated by the TGFβ1 via the ALK5 kinase in the PDAC line PANC-1 [14], the mutual inhibition of ALK5 and RAC1b constitutes a double-negative feedback loop. Although the downregulation of the *TGFBR1* by RAC1b—and, as a consequence, the reduced ALK5-dependent activation of ALK2—is likely to be effective in blocking TGFβ signaling to SMAD1/5, the ALK5-independent negative regulation of ALK2 by RAC1b, if operating, would make this inhibition even more efficient. We, therefore, tested the prediction that *ACVR1*, too, is targeted by RAC1b for inhibition.

We have recently established the PDAC-derived cell line PANC-1 as a suitable model to study EMTs/METs with a particular focus on the control of migration/invasion. Originally classified as quasimesenchymal [15], we were able to show that this cell line is heterogeneous in composition, consisting of cells with both mixed and mesenchymal phenotypes [14]. Induction of the EMT in these cells by the TGFβ1 was associated with a strong increase in migratory activity, while the MET induction was evident by a concomitant decrease [14]. Concomitantly, we observed a downregulation of the mesenchymal markers with promigratory functions, such as RAC1, vimentin, and SNAIL, and an upregulation of the epithelial markers, such as Claudin-4, GRHL2, and OVOL2, most of which also represent TGFβ target genes [14].

The role of ALK2 in the MET’s induction (via BMP7) or EMT inhibition, which proceeds in part by antagonizing TGFβ signaling [3,4,5,6,8,9,16], implies that ALK2 may also interfere with cell motility; however, the recent demonstration of ALK2 activation by TGFβ-bound ALK5 and subsequent ALK2-induced SMAD1/5 signaling was required for the EMT’s induction [11] somehow contradicts this assumption. In the latter study, TGFβ-induced Smad1/5 activation was not required for the growth-inhibitory effects of the TGFβ but was specifically required for TGFβ-induced anchorage-independent growth [12]; however, its impact on cell motility has not been determined. Here, we employed a combination of pharmacological, genetic, and dominant-negative inhibition strategies to selectively interfere with the ALK2 function in a battery of established and primary PDAC cell lines. We were able to show that ALK2 is a potent suppressor of basal and TGFβ1-induced migratory activity by establishing autoregulatory negative feedback with promigratory RAC1 proteins. In addition, we found that the protective effect of ALK2 against basal and TGFβ-dependent migration was stronger in the PDAC cells of the epithelial subtype as opposed to those presenting with a mixed epithelial–mesenchymal or mesenchymal (also termed quasimesenchymal) phenotype.

## 2. Materials and Methods

### 2.1. Cell Lines and Transient Transfections

Three PDAC-derived permanent cell lines of different transcriptional subtypes were employed, COLO 357 (classical/epithelial), PANC-1 (quasimesenchymal/mixed), and IMIM-PC-1 (mesenchymal), as well as the breast cancer cell line MDA-MB-231 (mesenchymal), were all maintained in an RPMI-1640 medium supplemented with 10% (*v*/*v*) fetal bovine serum (FBS), L-glutamine, sodium pyruvate and 1% penicillin/streptomycin (P/S) (all from Thermo Fischer Scientific, Darmstadt, Germany). A primary cell line, LuePanc-1, was derived from a pT3,pN1,G2 tumor, as described in detail elsewhere [17] and cultivated in a DMEM GlutaMAX^TM^ (Thermo Fisher Scientific, Darmstadt, Germany) with 4.5 g/L of glucose (Merck, Darmstadt, Germany) supplemented with 20% FBS and 1% P/S [17]. The transient transfection of PANC-1 cells with small interfering RNA (siRNA) or plasmid vectors, as well as the generation and characterization of PANC-1 cells with a genomic knockout (KO) of exon 3b of *RAC1*, has been published previously [13]. In the ALK2 siRNA and expression vector experiments, the mock-transfected samples did not differ from control siRNA and empty vector samples, respectively, and were, therefore, omitted from the respective graphs in Figure 1.

### 2.2. Immunoblotting

Cell lysis and immunoblotting were essentially performed as described previously [13,18,19] with minor modifications. Proteins were fractionated by polyacrylamide gel electrophoresis on TGX Stain-Free FastCast gels (BioRad, Munich, Germany). Following blotting and antibody treatment, chemoluminescent detection of proteins was done on a ChemiDoc XRS imaging system (BioRad, Munich, Germany) with an Amersham ECL Prime Detection Reagent (GE Healthcare, Munich, Germany). The antibodies used were: ALK2: Biorbyt Ltd., Cambridge, UK; phospho-SMAD1/5 and SMAD1: Cell Signaling Technology (CST), Frankfurt/Main, Germany; RUNX3/AML2: D9K6L, CST; RAC1: #610650, BD Biosciences, Heidelberg, Germany; RAC1b: #09-271, Merck Millipore, Darmstadt, Germany; GAPDH: 14C10, CST; HSP90: #13119, Santa Cruz Biotechnology, Heidelberg, Germany. The signals for the proteins of interest were normalized to the total amount of protein in the same lane, and significant differences (*p* < 0.05) were calculated with the unpaired two-tailed Student’s *t* test.

### 2.3. Quantitative PCR Analysis

The procedure and the conditions for real-time quantitative RT-PCR (qPCR), which was performed on an I-cycler with IQ software (Bio-Rad, Munich, Germany), were described in detail earlier [13,18,19]. The single-stranded cDNA was directly used for the quantitative real-time PCR. For amplification of the genes of interest and in separate reactions of the housekeeping gene, a TATA box-binding protein (TBP) was analyzed with a Maxima SYBR Green qPCR master mix (Thermo Fischer Scientific; Cat #: K0222), according to the manufacturer’s protocol (BioRad, Munich, Germany), and relative expression was calculated according to the delta Ct method.

### 2.4. xCELLigence-Based Cell Migration Assays

We employed the xCELLigence^®^ DP system (ACEA Biosciences, San Diego, CA, USA) to measure random/spontaneous cell migration in a chemokinesis setup according to previous descriptions [13,14,18]. Each well of the CIM plates-16 received 60,000–80,000 cells in a standard growth medium supplemented with 1% FBS. Data acquisition was done at intervals of 15 min and the assays were run for various lengths of time and analyzed with RTCA software (version 1.2, ACEA Biosciences).

### 2.5. Statistical Analysis

All statistical analyses were performed using either two-tailed unpaired Student’s *t* tests or a Mann–Whitney U test. *p* values of <0.05 were considered significant.

## 3. Results

### 3.1. Inhibition of the BMP Type I Receptor ALK2 Enhances Basal and TGFβ1-Induced Cell Migration

Given the proposed role as an epithelial marker and signaling intermediate of the MET inducer and TGFβ antagonist BMP7, we reasoned that ALK2 should negatively impact the EMT-associated functions, such as cell motility. To verify this experimentally, we silenced the ALK2 protein expression in the quasimesenchymal PDAC cell line, PANC-1, via the ALK2 RNA interference (RNAi)-mediated knockdown (KD) (Figure 1A) or inhibited the ALK2 function by a dominant-negative interference through the transient ectopic expression of a kinase-dead mutant, ALK2-K233R (ALK2-KR, Figure 1B) and, subsequently, subjected the cells to real-time cell migration assays. To this end, both approaches enhanced the basal and TGFβ1-induced cell migratory activities (Figure 1A,B). These data clearly show that in contrast to ALK5, ALK2 is a potent inhibitor of basal and TGFβ1-driven cell migration in PDAC-derived cells.

The small GTPase RAC1 is known for being a strong mediator of the TGFβ-induced EMT, and EMT-associated functions, such as stem cell generation and cell migration/invasion in PDAC and other tumor types [20,21]. Consistent with this, a decrease in RAC1 protein abundance has been noted earlier in PANC-1 cells upon the MET’s induction with a newly identified combination of TNFα, IL1β, and IFNɣ [14]. This suggested the possibility that RAC1 may also be downregulated in the course of the BMP7/ALK2-induced MET and that RAC1 is a (direct) target of ALK2 signaling. To this end, in cells ectopically expressing ALK2-KR, we observed an upregulation of RAC1, as well as in the *RAC1* splice isoform, RAC1b, as measured by immunoblotting (Figure 1C). Both RAC1 isoforms are expressed in PANC-1 cells and in other PDAC-derived cell lines, as analyzed in Figure 1, Appendix A and [19]. From these data, we conclude that ALK2 is a negative regulator of cell migration and that this effect is mediated, at least in part, through the downregulation of RAC1 and/or RAC1b.

### 3.2. ALK2 Expression, SMAD1/5 Signaling, and Target Gene Expression Is Subject to Negative Regulation by RAC1b

Above, we have shown that ALK2 represses RAC1 and RAC1b, which have both been implicated in driving invasion, metastasis, and pancreatic cancer progression [20,21,22,23], providing a mechanistic basis for ALK2′s antimigratory activity. Since EMT/MET regulation often involves negative feedback loops, i.e., between the transcription factors ZEB1/2 and the microRNA miR-200 [24], we speculated that such a loop may also operate in the ALK2-dependent control of the MET. More specifically, we asked if RAC1b would be capable of suppressing ALK2. To address this issue, we selectively inhibited RAC1b in PANC-1 cells by an RNAi-mediated KD or genomic KO via CRISPR/Cas9 technology and determined the effect on ALK2 mRNA expression. With both gene silencing strategies, increased mRNA levels of ALK2, but not ALK3 or ALK6, were observed (Figure 2A,B). ALK2 protein levels were also found to be increased in RAC1b-KO cells (Figure 2C). Of note, under conditions of RAC1b silencing, the exogenous TGFβ1 was more potent in up-regulating ALK2 expression (Figure 2A). Finally, we asked if the derepression of ALK2 following RAC1b silencing would also impact ALK2-induced signaling—following the TGFβ1 stimulation—known to proceed through SMAD1/5 rather than SMAD2/3. To this end, in RAC1b-deficient PANC-1 cells, we observed a dramatic increase in levels of C-terminally phosphorylated SMAD1/5 proteins relative to the lentiviral (LV) control cells (Figure 2D).

The Runt-related transcription factor 3 (RUNX3) is involved in the MET’s induction by decreasing the sensitivity to the TGF-β1-induced EMT and reversing the EMT through the TGFβ/Smad signaling and by inhibiting the invasion and migration of esophageal squamous cell carcinoma (ESCC) cells [25]. RUNX3 was chosen here for further analysis because unlike other MET and EMT markers it is also a promoter of BMP9-ALK2-pSMAD1/5/8 signaling [26]. Consistent with the inhibitory effect on both ALK2 expression and SMAD1/5 activation, RAC1b also appears to inhibit RUNX3, as revealed by a dramatic induction of RUNX3 protein levels in the PANC-1-RAC1b-KO cells (Figure 2E). This suggests that RAC1b, in addition to suppressing ALK5-SMAD3 signaling [13], negatively controls the TGFβ signaling via the ALK2-SMAD1/5 arm.

### 3.3. The Inhibitory Effect of ALK2 on TGFβ-Driven Cell Migration Depends on the PDAC Differentiation Subtype

PDACs are classified according to histomorphological phenotypes and transcriptionally defined subtypes as either classical/epithelial, purely mesenchymal, or quasimesenchymal [15]. Based on the generally low or absent invasive activity in well-differentiated tumor cells (as opposed to the highly invasive poorly differentiated ones), we reasoned that ALK2 may have a more prominent role in suppressing TGFβ-induced migration in epithelial subtype cells as opposed to highly mesenchymal subtype cells. To analyze this in more detail, we performed a series of migration assays with four SMAD4-wildtype PDAC cell lines and the BMP type I receptor inhibitor, LDN-193189 [11]. LDN-193189 is a selective BMP signaling inhibitor that inhibits the kinase activity of ALK2 and the related ALK1 and at a concentration of 1 μM has been shown to also partially and selectively inhibit TGFβ-induced SMAD1/5C phosphorylation in the breast cancer cell line, MDA-MB-231 [11]. Since the expression of ALK1 is restricted to endothelial cells [27], we reasoned that 1 μM of LDN-193189 specifically targets ALK2 in our cells. Further, given the antimigratory effect of ALK2 demonstrated after RNAi and the genomic KO-mediated silencing of *ACVR1* in the PANC-1 cells (Figure 1), we reasoned that the pharmacologic inhibition of ALK2 with LDN-193189 should result in a relative preponderance of ALK5-SMAD2/3 signaling and a concomitant *increase* in migratory activity. To this end, we, indeed, observed this increase in 3/4 of the cell lines and, strikingly, the extent of the stimulatory effect of LDN-193189 on TGFβ-driven cell invasion correlated with the grade of the epithelial nature: strong in COLO 357 (classical/epithelial), moderate in PANC-1 (mixed/quasimesenchymal), weak in IMIM-PC-1 (mesenchymal), and absent in MDA-MB-231 cells (extremely mesenchymal) (Figure 3A). In contrast, the ALK5 inhibitor, SB-431542, which blocks conventional ALK5-SMAD2/3 signaling [28], inhibited cell migration in the COLO 357, PANC-1, and IMIM-PC-1 cells (the MDA-MB-231 cells were not tested) (Figure 3A).

Primary cancer cell cultures established in our laboratory from surgically resected PDAC tissue yielded LuePanc-1 cells [17]. These cells have been partially characterized earlier to display the readily detectable expression of both epithelial markers, i.e., E-cadherin and mesenchymal markers, i.e., vimentin, as well as of RAC1 and RAC1b (Appendix A), indicating a mixed phenotype [29]. This classification predicted only a moderate inhibitory effect of ALK2 signaling on migratory activity. Indeed, the treatment of these cells with 1 µM of LDN-193189 + TGFβ1 resulted in a small but statistically significant increase in the migratory activity over the TGFβ1 alone (Figure 3B, blue curve vs. red curve) and the extent of this increase resembled that in the PANC-1 cell line (Figure 3A).

These data indicate the presence of two TGFβ-stimulated type I receptors and their respective signaling pathways and suggest a functional antagonism between them in the control of cell migration. Moreover, differences in TGFβ-ALK2-SMAD1/5 signaling activity appear to exist among the PDAC cells representing different tumor subtypes. The phenomenon of the antagonistic regulation of cell invasion by the same ligand (TGFβ) in the same cell via the employment of two different members of type I receptors (ALK5 and ALK2) and their SMAD targets (SMAD2/3 and SMAD1/5, respectively) is intriguing.

## 4. Discussion

The goal of this study was to evaluate the role of the BMP type I receptor ALK2 in cell migration in vitro of PDAC-derived cells. Our data suggest that (i) ALK2 inhibits migration by suppressing RAC1/RAC1b protein expression, (ii) ALK2 and RAC1b act together in a negative feedback loop to mutually control their expression and signaling, and (iii) the ALK2 antimigratory function appears to be particularly crucial in protecting epithelial subtype cells from becoming invasive, both spontaneously and in a TGFβ-rich tumor microenvironment.

The available studies on ALK2 have assessed its effect on the EMT but did not provide data on specific measurements of cell migration in vitro. Our results here are, therefore, the first ones to reveal an inhibitory role in both spontaneous and TGFβ-driven invasion using PDAC-derived cells as a model system. TGFβ1-induced SMAD1/5 phosphorylation initially involves the ligand-induced phosphorylation of ALK5, which again by phosphorylation, subsequently activates ALK2, which in turn phosphorylates SMAD1/5. SMAD1/5 signaling in response to the TGFβ has been shown to be required for a complete TGFβ-induced EMT in NMuMG and EpRas cells and approximately a quarter of the TGFβ-induced transcriptome was found to depend on SMAD1/5 signaling. This accounts for a previously unexplained observation, namely that the overexpression of dominant-negative ACVR1 in NMuMGs caused a partial loss of the EMT in response to the TGFβ [11,30]. However, unfortunately, in the latter study, it was not tested whether this incomplete EMT was associated with reduced cell motility [30]. Our data clearly support such a scenario. The anti-migratory function of ALK2 would also be in support of its anti-EMT/pro-MET function, as discussed below.

Having identified ALK2 as a negative regulator of cell migration, we were keen to identify the downstream targets of ALK2 that may mediate the antimigratory effect. Using an immunoblot analysis of PANC-1 cells expressing a kinase-dead version of ALK2, we observed the derepression of RAC1 and the *RAC1* splice isoform, RAC1b, and have, thus, tentatively identified RAC1 and RAC1b as negative targets of ALK2 signaling in PDAC cells. Since both RAC1 [21] and RAC1b [21,22,23] have been implicated in driving migration/invasion and malignant progression in several carcinomas, including PDAC, it is suggested that ALK2, in order to suppress cell migration and invasion, downregulates the abundance of the RAC1 and RAC1b proteins.

Regulators of central developmental processes such as EMT/MET and EMT-associated functions often act in the form of negative feedback loops. Consequently, we addressed the question if RAC1b can also control the expression of ALK2 in a negative fashion and much to our surprise, we found this to be the case. The RAC1b regulation of ALK2 appears to operate at the transcriptional level since both the ALK2 protein and mRNA levels were similarly increased in PANC-1 cells in response to the RAC1b-KD or KO. We concluded that RAC1b targets ALK2 directly for inhibition and that this regulatory effect of RAC1b was specific for ALK2 since the mRNA levels of the related BMP type I receptors, ALK3 or ALK6, were not affected.

As ALK2 signals through SMAD1/5 in response to either BMP7 or the TGFβ [5,6,7,8,9], we further investigated if RAC1b-KO impacts the phosphorylation state of these Smads. Intriguingly, the abundance of C-terminally phosphorylated SMAD1/5 in PANC-1-RAC1b-KO cells was dramatically elevated after the TGFβ1 stimulation (Figure 2D). Since we have previously shown that RAC1b also efficiently suppresses ALK5 mRNA and protein levels [13], this dramatic derepression of activated SMAD1/5 may have resulted from the combined independent suppression of ALK5 and ALK2, both of which are required for SMAD1/5 activation [11]. Further proof that the derepression of ALK2 is involved here came from transfection experiments with kinase-dead ALK2-KR, which rescued RAC1b-KO/KD cells from the TGFβ1-induced increase in phospho-SMAD1/5C levels (the data are not shown).

In an analogy to the negative regulation of RAC1b expression by ALK2, we demonstrated earlier that RAC1b is also subject to negative regulation by the TGFβ1 via the ALK5 kinase [14]. From this, it appears that the TGFβ1/ALK5-induced downregulation of RAC1b involves ALK2-SMAD1/5 signaling. Indeed, LDN-193189 at 1 µM partially inhibited the TGFβ1-induced downregulation of RAC1b protein levels (H.U., unpublished observation). Taken together, our data are in favor of a self-perpetuating autoregulatory feedback loop between RAC1b and ALK2, in which RAC1b suppresses ALK2 and ALK2-induced SMAD1/5 signaling, and, conversely, ALK2 suppresses RAC1b protein expression (Figure 4).

Very little is known about the possible nuclear target genes downstream of ALK2-SMAD1/5-RAC1b to impact cell invasion. For instance, in the TGF-β-induced EMT, SMAD1/5 signaling was essential for the induction of Inhibitor of Differentiation 1 (ID1) [11]. RUNX3 has been identified as another ALK2-SMAD1/5-SMAD4 target gene, which is also supported by our finding that in RAC1b-KO cells exhibiting elevated ALK2 and pSMAD1/5 levels, RUNX3, too, was dramatically upregulated (Figure 2E). RUNX3 may be involved, here, in blocking or reversing the TGFβ1-induced EMT, as RUNX3 overexpressing cells displayed less sensitivity to the TGFβ1-induced EMT [25]. Moreover, RUNX3 inhibited the invasion and migration of ESCC cells by reversing the EMT through TGFβ/Smad signaling, consistent with RUNX3 overexpression markedly inhibiting the phosphorylation of SMAD2/3 [25]. The functional operation of an ALK2-SMAD1/5-SMAD4-RUNX3 axis activated, i.e., by BMP7, which counteracts the canonical ALK5-SMAD2/3-SMAD4 signaling, may underly the opposing roles and potential antagonistic mechanisms between the BMP7 and TGFβ pathways in cancer [3,5,6]. This may mean that the higher TGFβ concentrations (0.5–5 ng/mL) normally used in cell culture experiments activate both pathways, which are functionally antagonistic to cell motility and that, by selectively inhibiting ALK2-SMAD1/5 by 1 μM of LDN-193189, results in a preponderance of ALK5-SMAD2/3 signaling and, hence, an increase in TGFβ1-induced migration. This is supported by the effective inhibition of ALK5-SMAD2/3-dependent migratory activity by SB-431542. Both Smad pathways may even operate downstream of the same ligand, i.e., the TGFβ, as the selective inhibition of ALK2 kinase activity with LDN-193189 would remove an invasion constraint from the cells to shift the balance from the ALK2-SMAD1/5 to the ALK5-SMAD2/3 branch and, therefore, increased migratory activity. In support of this scenario, we observed, in several PDAC cell lines, the derepression of the basal invasive activity upon the inhibition of ALK2-SMAD1/5 activation by ALK2-KR (in PANC-1) and LDN-193189 (in COLO 357, PANC-1, IMIM-PC-1, and LuePanc-1 cells). Moreover, PDAC cells representing different tumor subtypes exhibited different degrees of sensitivity to pharmacological ALK2 inhibition with respect to TGFβ-driven cell invasion, with classical/epithelial COLO 357 cells being the most, and mesenchymal IMIM-PC-1 and MDA-MB-231 cells being the least, responsive. The molecular basis for these differences in the sensitivity to the LDN-193189-mediated inhibition of ALK2 is currently not clear but may involve differences in the abundance of the ALK2 or SMAD1/5 proteins, or the further downstream located signaling intermediates for which the ALK2-SMAD1/5 and ALK5-SMAD2/3 branches compete in order to be able to transmit the signal to the nucleus, i.e., e.g., RAC1, RAC1b, or SMAD4.

BMP7 acting via ALK2 and SMAD1/5 has been shown to induce the MET in melanoma cells and inhibit metastasis [8]. Rather than blocking the TGFβ ligand or ALK5, selective promotion of the ALK2-SMAD1/5 arm of TGFβ signaling could be exploited for antimetastatic therapy without compromising the trophic ALK5-mediated functions of this growth factor.

We have previously identified RAC1b as a gate-keeper of the epithelial phenotype [19] and an inhibitor of the EMT. Since TGFβ-induced EMT required signaling via both the activation of ALK5 (with subsequent SMAD3-dependent signaling) and the ALK5-mediated activation of ALK2 (with subsequent SMAD1/5-dependent signaling) [11,30], the RAC1b-mediated independent inhibition of both ALK5 [13] and ALK2 (this study) expression and signaling may form the mechanistic basis of this strong anti-EMT effect.

## Figures and Tables

**Figure 1 biomedicines-10-02640-f001:**
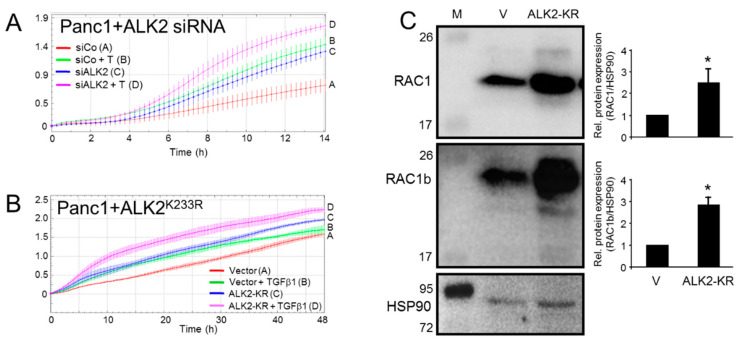
Selective inhibition of the ALK2 kinase function enhances basal and TGFβ1-dependent cell migration. (**A**) PANC-1 cells were transiently transfected with either an ALK2-specific siRNA (siALK2) or an irrelevant control siRNA (siCo). Forty-eight hours after the start of transfection, cells were subjected to a real-time cell migration assay on an xCELLigence platform in the absence or presence of 5 ng/ml rec. human TGFβ1 (T). Following the addition of the TGFβ1, cells were immediately assayed for the presence of migratory activity. (**B**) As in (**A**), except that those cells received a kinase-dead ALK2 mutant (K233R mutation, ALK2-KR) or empty pcDNA3 vector and were treated, or not, with the TGFβ1 during the assay. The data shown in (**A**,**B**) are representative of three independent assays performed in total. (**C**) Immunoblot analysis of PANC-1 cells ectopically expressing either an empty vector (V) or ALK2-KR. HSP90 was detected on the same blots to verify equal loading. M, molecular weight marker; the numbers on the left-hand side indicate the sizes of the marker bands in kDa. The asterisks (*) indicate a significant difference.

**Figure 2 biomedicines-10-02640-f002:**
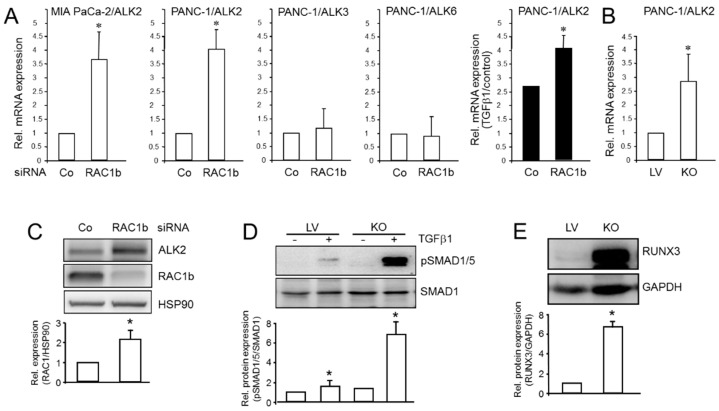
RAC1b negatively regulates ALK2 expression and signaling in response to the TGFβ stimulation in PANC-1 cells. (**A**,**B**) Impact of RAC1b KD (**A**) or KO (**B**) on ALK2 mRNA expression. The most right-handed graph with black-filled bars in (**A**) shows the effect of the TGFβ1 on ALK2 mRNA abundance in RAC1b-KD cells displayed as –fold induction by the TGFβ1 over the control. (**C**) Impact of RAC1b siRNA on ALK2 protein expression. Lysates from PANC-1 cells transfected with either an irrelevant control siRNA (Co) or a RAC1b-specific siRNA were sequentially analyzed by immunoblotting for ALK2, RAC1b, and HSP90 as a loading control. (**D**) SMAD1/5 activation in PANC-1-RAC1b-KO (KO) and LV control cells in response to the TGFβ1 treatment (1 h) as assessed by immunoblotting for C-terminally phosphorylated SMAD1/5 (pSMAD1/5). (**E**) Immunoblot analysis of RUNX3 in PANC-1-RAC1b-KO cells. PANC1-RAC1b-KO and LV control cells were subjected to immunoblot analysis and sequentially probed with antibodies to RUNX3 and GAPDH as a loading control. The graphs in C-E represent the densitometric readings from three independent experiments (mean ± SD). The blot shown is representative of three experiments. The asterisks (*) indicate a significant difference.

**Figure 3 biomedicines-10-02640-f003:**
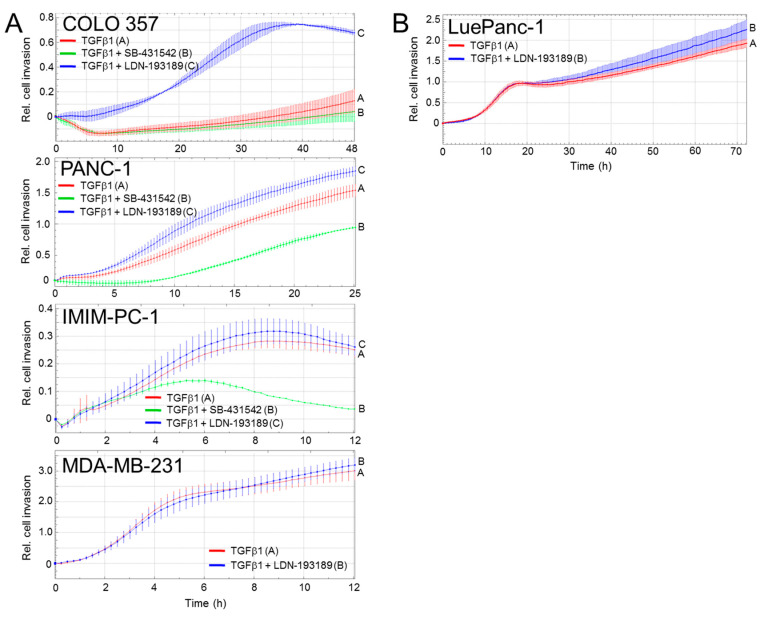
The inhibitory effect of ALK2 on basal and TGFβ-induced migration depends on the PDAC differentiation subtype. (**A**) Real-time cell migration assays with COLO 357, PANC-1, IMIM-PC-1 cells and MDA-MB-231 cells, in the presence or absence of the TGFβ1 (5 ng/ml), LDN-193189 (1 μM), or SB-431542 (5 µM), as indicated. (**B**) As in (**A**), except that the primary cell line, LuePanc-1 was used and treated, or not, with LDN-193189. Each assay in (**A**,**B**) is representative of at least three independent assays performed in total.

**Figure 4 biomedicines-10-02640-f004:**
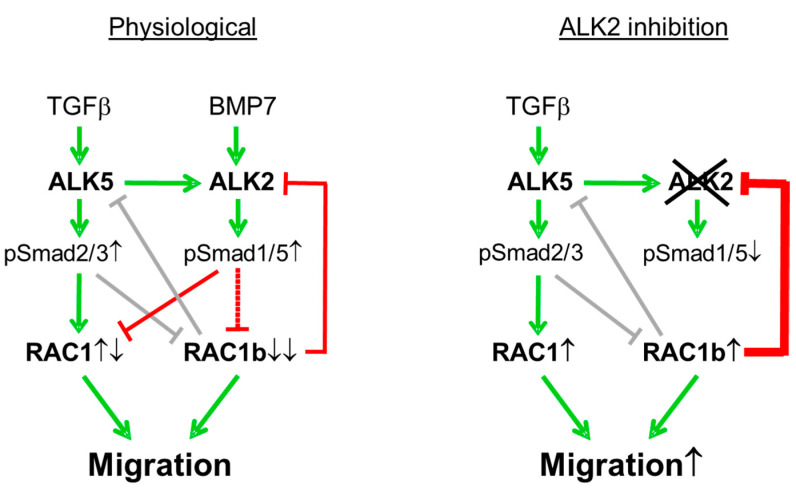
Cartoon summarizing the regulatory interactions between the type I receptors ALK5 and ALK2, and the small GTPases RAC1 and RAC1b, in pancreatic tumor cells under physiological conditions (**left-hand side**) and after selective inhibition of the ALK2 kinase (**right-hand side**). Green arrows denote stimulatory interactions (induction of expression and/or activity) and red lines indicate inhibitory ones. Gray-shaded lines denote a previously identified, self-perpetuating autoregulatory feedback loop between ALK5-SMAD2/3 and RAC1b. Under physiological conditions, RAC1 receives positive and negative inputs from ALK5-SMAD2/3 and ALK2-SMAD1/5, respectively, while RAC1b receives negative input from both receptors, eventually leading to its strong suppression (↓↓) in response to ligand-induced receptor activation. Following ALK2 inhibition with either RNAi, dominant negative interference, or LDN-193189 treatment, the formation of phosphorylated SMAD1/5 (pSmad1/5) is reduced (↓), resulting in derepression (↑) of both RAC1 and RAC1b and, consequently, increased migratory activity. The elevated amount of RAC1b protein accumulating in response to forced inhibition of ALK2 may aid in keeping cellular ALK2 levels low (bold red line).

## Data Availability

All data are contained within the main figures and the Appendix A.

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
