# Peer review of "Suppressive Role of ACVR1/ALK2 in Basal and TGFβ1-Induced Cell Migration in Pancreatic Ductal Adenocarcinoma Cells and Identification of a Self-Perpetuating Autoregulatory Loop Involving the Small GTPase RAC1b"

_biomedicines, 2022, doi:10.3390/biomedicines10102640_

Round 1
Reviewer 1 Report
The authors have examined the effects of ALK2 and RAC1b on the migration of PDAC cells using RNAi, dominant negative mutants, CRISPR, and chemical inhibitors, and found the suppressive role of ALK2 and RAC1b on the migration with a mutually negative feedback. The results are beautiful and the finding is novel and interesting.
The authors used 'double-negative feedback loop' between ALK2 and RAC1b. If ALK2 happens to increase a little bit, RAC1 decreased. Then, the decreased RAC1 shows less suppressive activity against ALK2. Then, ALK2 increases more and more, and RAC1 decreases less and less. The double-negative feedback loop means that either of ALK2 and RAC1 can work in a single cell. Thus, ALK2 and RAC1 suppressed with each other but not in a negative feedback. 'double-negative feedback loop' may not be the right term.
The authors wanted to examine the cell invasive activity by using only xCELLigence-based migration assay. It may be better to use some additional invasion assay to confirm it.
Although RUNX3 was examined as a MET marker, it may be better to use some additional markers for EMT and MET.
Discussion is well written but redundant. Even the response was variable among PDAC cell types, it may be no use to discuss on other cancer types.
The paucity of data in this manuscript may have led to long discussion. Some supportive experiments are expected.
Minor comments
Figure 1: Time schedule is not shown. How long cells were cultured after siRNA transfection? How long cells were pretreated with TGFβ1? Alternatively, cells were added with TGFβ1 and immediately assayed for the migration? The final concentration of TGFβ1 was 5 ng/ml ?
Line 289: 'transcriptional activity of ALK2' does not make sense. Does it mean the mRNA expression of ALK2, transactivation by ALK2, transactivation of the ALK2 gene, or the effects of AKL2 on the transcription ?
There are some errors in commas and periods.
Author Response
Dear Editor, dear Peerada Ngamsnae,
we would like to thank the reviewers for their enthusiastic comments on our manuscript and have done our best to incorporate their suggested changes and additions into the revised version. We believe that this has further improved its quality. All changes have been highlighted by the “track changes” mode. Please note that in response to a comment of Reviewer 1, the title has been changed slightly.
Reviewer 1
The authors have examined the effects of ALK2 and RAC1b on the migration of PDAC cells using RNAi, dominant negative mutants, CRISPR, and chemical inhibitors, and found the suppressive role of ALK2 and RAC1b on the migration with a mutually negative feedback. The results are beautiful and the finding is novel and interesting.
1) The authors used 'double-negative feedback loop' between ALK2 and RAC1b. If ALK2 happens to increase a little bit, RAC1 decreased. Then, the decreased RAC1 shows less suppressive activity against ALK2. Then, ALK2 increases more and more, and RAC1 decreases less and less. The double-negative feedback loop means that either of ALK2 and RAC1 can work in a single cell. Thus, ALK2 and RAC1 suppressed with each other but not in a negative feedback. 'double-negative feedback loop' may not be the right term.
Response: We agree with the reviewer and have replaced the term 'double-negative feedback loop' by the term “self-perpetuating autoregulatory feedback loop”.
2) The authors wanted to examine the cell invasive activity by using only xCELLigence-based migration assay. It may be better to use some additional invasion assay to confirm it.
Response: We apologize for confusing the terms invasion and migration. With the setup chosen for our experiments we wanted to measure cell migration. The reviewer is correct in that we have not employed the invasion setup for this assay. Therefore, we have replaced the term “invasion” by “migration” throughout the manuscript.
3) Although RUNX3 was examined as a MET marker, it may be better to use some additional markers for EMT and MET.
Response: The regulation of several MET and EMT markers by RAC1b was shown in previous publications (Ref. 14 and 19, respectively). RUNX3 was chosen here because it is not only a marker of MET but also a promoter of BMP9-ALK2-pSMAD1/5/8 signaling (Ref. 26). For other MET and EMT markers it is not known whether they can promote BMP9-ALK2-pSMAD1/5/8 signaling. We have added an explanatory sentence to the second paragraph of section 3.2.
4) Discussion is well written but redundant. Even the response was variable among PDAC cell types, it may be no use to discuss on other cancer types.
Response: As requested, we have shortened the discussion and have removed the last six lines. However, we believe that further shortening would remove important aspects and pieces of information that could aid in evaluating the implications and perspectives of our findings.
5) The paucity of data in this manuscript may have led to long discussion. Some supportive experiments are expected.
Response: We have added qPCR data on ALK2 expression in RAC1b-silenced cells stimulated or not with TGFb1. Besides the identification of ALK2 as a TGFβ target gene, we found that in RAC1b silenced cells, TGFβ1 was more potent in up-regulating ALK2 mRNA expression. These data have been included as the most right-handed graph with black-filled bars in Figure 2A and their description in the first paragraph of section 3.2.
Minor comments
1) Figure 1: Time schedule is not shown. How long cells were cultured after siRNA transfection? How long cells were pretreated with TGFβ1? Alternatively, cells were added with TGFβ1 and immediately assayed for the migration? The final concentration of TGFβ1 was 5 ng/ml ?
Response: Cells were cultured for 48 hours after the start of siRNA transfection. Following addition of TGFβ1, cells were immediately assayed for the migration. Yes, the final concentration of TGFβ1 was 5 ng/ml. All these pieces of information were given now in the legend to Figure 1.
2) Line 289: 'transcriptional activity of ALK2' does not make sense. Does it mean the mRNA expression of ALK2, transactivation by ALK2, transactivation of the ALK2 gene, or the effects of AKL2 on the transcription ?
Response: This error has been removed and corrected by replacing the term “transcriptional activity” by “kinase activity”.
3) There are some errors in commas and periods.
Response: These have been rectified.
Reviewer 2 Report
The experiments presented in this manuscript were designed to analyze the role of TGFbeta-receptor ALK2 in the regulation of pancreatic ductal adenocarcinoma cell migration. In addition, the behaviour of several elements (RAC1, RAC1B, SMAD1/5, RUNX3) of the highly complex signaling mechanisms controling cell movements and cancer cell invasion were studied. To this end, several methods to inhibit ALK2 (siRNA knockdown, expression of a dominant negative ALK2, pharmacological inhibition of the enzyme) and RAC1B (knockdown and CRISPR-KO) had been used. The results are clearly described and interpreted, they strongly support the conclusions of the authors listed in the first paragraph of Discussion. (Some details of this signaling network need further investigation, e.g. the controversial role of RAC1B in the invasion process.)
A minor comment: in the siRNA and ALK2-K233R experiments no-transfection samples are not shown. If these did not differ from control siRNA and empty vector samples, respectively, this should be stated in Materials and Methods.
Author Response
Dear Editor, dear Peerada Ngamsnae,
we would like to thank the reviewers for their enthusiastic comments on our manuscript and have done our best to incorporate their suggested changes and additions into the revised version. We believe that this has further improved its quality. All changes have been highlighted by the “track changes” mode. Please note that in response to a comment of Reviewer 1, the title has been changed slightly.
Reviewer 2
The experiments presented in this manuscript were designed to analyze the role of TGFbeta-receptor ALK2 in the regulation of pancreatic ductal adenocarcinoma cell migration. In addition, the behaviour of several elements (RAC1, RAC1B, SMAD1/5, RUNX3) of the highly complex signaling mechanisms controling cell movements and cancer cell invasion were studied. To this end, several methods to inhibit ALK2 (siRNA knockdown, expression of a dominant negative ALK2, pharmacological inhibition of the enzyme) and RAC1B (knockdown and CRISPR-KO) had been used. The results are clearly described and interpreted, they strongly support the conclusions of the authors listed in the first paragraph of Discussion. (Some details of this signaling network need further investigation, e.g. the controversial role of RAC1B in the invasion process.)
A minor comment: in the siRNA and ALK2-K233R experiments no-transfection samples are not shown. If these did not differ from control siRNA and empty vector samples, respectively, this should be stated in Materials and Methods.
Response: The no-transfection/mock-transfection samples did indeed not differ from the control siRNA- or empty vector-transfected samples. This piece of information has been added to the Material and Methods section (2.1.), as requested.